

# Living in the intertidal: desiccation and shading reduce seagrass growth, but high salinity or population of origin have no additional effect

Wouter Suykerbuyk[1,2], Laura L. Govers[2,3,4], W.G. van Oven[1], Kris Giesen[1,2], Wim B.J.T. Giesen[5], Dick J. de Jong[6], Tjeerd J. Bouma[1] and Marieke M. van Katwijk[1,2]

[1] Department of Estuarine and Delta Systems, and Utrecht University, Royal Netherlands Institute for Sea Research (NIOZ), Yerseke, Netherlands
[2] Department of Environmental Science, Institute for Water and Wetland Research, Radboud University Nijmegen, Nijmegen, Netherlands
[3] Institute for Evolutionary Life Sciences (GELIFES), Conservation Ecology Group, University of Groningen, Groningen, Netherlands
[4] Department of Aquatic Ecology and Environmental Biology, Institute for Water and Wetland Research, Radboud University Nijmegen, Nijmegen, Netherlands
[5] Euroconsult Mott MacDonald, Arnhem, Netherlands
[6] Zee en Delta Department, Ministry of Infrastructure and Environment, Rijkswaterstaat, Middelburg, Netherlands

Corresponding author
Marieke M. van Katwijk,
m.vankatwijk@science.ru.nl

## ABSTRACT

The limiting effects of stressors like desiccation, light and salinity on seagrass growth and distribution are well-studied. However, little is known about their interactive effects, and whether such effects might differ among populations that are adapted to different local conditions. In two laboratory experiments we tested (a) if growth and development of intertidal, temperate *Zostera noltii* is affected by emergence time (experiment 1 and 2), and (b) how this is affected by an additional, second stressor, namely shading (experiment 1) or high salinity (25, 30 and 35, experiment 2). In addition, we tested (c) whether the effects of emergence time and salinity varied between three different European seagrass populations (Saint-Jacut/France, Oosterschelde/The Netherlands, and Sylt/Germany), which are likely adapted to different salinity levels (experiment 2). In both experiments, emergence of 8 h per tidal cycle (of 12 h) had a negative effect on seagrass relative growth rate (RGR), and aboveground biomass. Emergence furthermore reduced either rhizome length (experiment 1) or belowground biomass (experiment 2). Shading (experiment 1) resulted in lower RGR and a two-fold higher aboveground/belowground ratio. We found no interactive effects of emergence and shading stress. Salinity (experiment 2) did not affect seagrass growth or morphology of any of the three populations. The three tested populations differed greatly in morphology but showed no differential response to emergence or salinity level (experiment 2). Our results indicate that emergence time and shading show an additive negative effect (no synergistic or antagonistic effect), making the plants still vulnerable to such combination, a combination that may occur as a consequence of self-shading during emergence or resulting from algal cover. Emergence time likely determines the upper limit of *Z. noltii* and such shading will likely lower the upper limit.

Shading resulted in higher aboveground/belowground ratios as is a general response in seagrass. *Z. noltii* of different populations originating from salinity 30 and 35 seem tolerant to variations in salinity within the tested range. Our results indicate that the three tested populations show morphotypic rather than ecotypic variation, at least regarding the salinity and emergence, as there were no interactive effects with origin. For restoration, this implies that the salinity regime of the donor and receptor site of *Z. noltii* is of no concern within the salinity range 25–35.

## INTRODUCTION

Desiccation due to air exposure imposes a stress to marine life in the intertidal zone. Whereas mobile species can escape to moist places during low tide, sessile intertidal organisms need to cope with hours of air exposure. In the seagrass species *Zostera noltii*, short periods of air exposure are utilized to its advantage to assimilate $CO_2$, as long as the leaves remain moist (*Leuschner, Landwehr & Mehlig, 1998*). However, adverse effects of emergence rapidly increase with duration to air. Emergence periods of only 5 h air exposure per tide may result in a 50% water loss in the leaves, concomitant with 50% reduction in photosynthetic rates (*Leuschner, Landwehr & Mehlig, 1998*). Besides the physiological effects of drought stress, desiccation of the leaves after low tide exposure decreases the mechanical strength and subsequently the probability of leaf sloughing (*Vermaat et al., 1993*). In addition, this may result in shorter leaf lengths as desiccated leaf points are prone to break, resulting in a decreased capacity of photosynthesis (*Boese et al., 2003*), and reduced water retention by the leaves (*Fox, 1996*), which is density dependent (*De Fouw et al., 2016*).

Desiccation is usually the limiting factor controlling the upper limit of seagrass growth on the intertidal flat (*Philippart & Dijkema, 1995*; *Leuschner, Landwehr & Mehlig, 1998*; *Van Katwijk & Hermus, 2000*; *Van der Heide et al., 2010*). However, risk of desiccation varies over tides, days, seasons and latitude (*Perez-Llorens & Niell, 1993*), and actual emergence stress will depend on the temperature, wind conditions, sediment water content and seagrass density. For example, although intertidal *Z. noltii* in the sub-tropical Mauretania has to cope with an air temperature of 40 °C during 6 h of emergence per tide, meadows still have high (>75% coverage) shoot density due to the facilitating effects of such high densities on water retention of meadows (*De Fouw et al., 2016*). On the other hand, the productivity during emersion is lower than during submergence in these beds (*Clavier et al., 2011*), which was also found in more temperate beds at the Atlantic coast of France (*Ouisse, Migné & Davoult, 2011*). In both cases, this was largely attributed to self-shading by the leaves laying flat, covering each other in the dense beds. It may seem counterintuitive that light could be limiting in an intertidal bed, but in addition to self-shading, very turbid water may also limit productivity during the high tide hours, and cover by epiphytes and green macroalgae can be severe in intertidal beds, particularly

since seagrasses facilitate the presence of these algae by providing substrate and shelter (*Michael et al., 2008*).

In temperate regions, light availability varies over the year and controls the start and end of rhizome branching during the growing season (*Vermaat & Verhagen, 1996*; *Govers et al., 2015*). Photo-inhibition is not likely, as *Z. noltii* is very tolerant to high light levels (*Jimenez, Niell & Algarra, 1987*) and also in combination with emergence (*Clavier et al., 2011*; *Ouisse, Migné & Davoult, 2011*). To compensate for light limitation, shading typically results in an increased aboveground/belowground biomass ratio with longer shoots compared to plants grown in ambient light conditions (*Abal et al., 1994*; *Philippart, 1995*; *Vermaat, Verhagen & Lindenburg, 2000*; *Peralta et al., 2002*). However, longer shoots may make intertidal seagrass plants more vulnerable to desiccation when growing at low density.

In general, several stressors may cause a conditional outcome of emergence stress. For example, in addition to light limitation due to self-shading or algal cover mentioned above, a stress like a high salinity may influence the tolerance to desiccation. Salinity stress could be one of the most ubiquitous stressors that marine life and thus also seagrasses encounter. Marine macrophytes can counteract osmotic stress on the short-term by internal adjustments of turgor-pressure by up- and down-regulation of simple ions and on the long-term by synthesis or breakdown of osmotically active compounds (*Touchette, 2007*). Both physiological mechanisms require energy and could therefore reduce plant fitness. A salinity range as wide as 10–35 does not cause increased mortality in seagrass *Zostera marina* (*Kamermans, Hemminga & de Jong, 1999*; *Van Katwijk et al., 1999*; *Nejrup, Brammer & Pedersen, 2008*), whereas salinities of five and lower increase mortality (*Nejrup, Brammer & Pedersen, 2008*). With increasing temperature, salinity stress effects become even more pronounced (*Salo & Pedersen, 2014*). Within the non-mortal salinity conditions, a salinity range of 22–23 was found to be the optimal ex situ for temperate eelgrass *Z. marina*. At this salinity, maximum production of shoots and leaves was found whereas growth and vitality were reduced at higher salinities (>26) (*Kamermans, Hemminga & de Jong, 1999*; *Van Katwijk et al., 1999*). Plants of *Z. marina* populations grown under high salinity seem to better cope with high salinity than plants originating from an estuarine or other low salinity habitat (*Van Katwijk et al., 1998*, *1999*), and reversely, low-salinity grown plants tolerate lower salinities than high salinity grown plants (*Salo, Pedersen & Bostrom, 2014*). Also, in *Posidonia oceanica*, distinct differences in response to salinity between plants from different origins are found (*Fernandez-Torquemada & Sanchez-Lizaso, 2005*). In *Z. noltii*, different origins were tested on high salinity tolerance, comparing 15 and 35 (*Vermaat, Verhagen & Lindenburg, 2000*). Responses to salinity stress are similar to those of *Z. marina*: increased mortality was found at extreme salinities (*Vermaat, Verhagen & Lindenburg, 2000*; *Charpentier et al., 2005*). To our knowledge, the more subtle salinity preferences of *Z. marina* plants of different origins described above were never tested for *Z. noltii*. The outcome of such tests can be important for restoration projects with different donors originating from different salinities. Since critical mass is very important for seagrass restoration success

(*Van Katwijk et al., 2016*), focus should not only be on *surviving* certain salinities but also about highest growth rates.

Although separate effects of desiccation, light and salinity stress are relatively well studied, little is known about the interactive effects of these stressors, in other words, whether they show an additive effect (= no interaction), or may act synergistic or antagonistic (= interaction), and if these effects differ among populations that are adapted or acclimatized to different local conditions (but see *Vermaat, Verhagen & Lindenburg, 2000*). Thus, we want to test if the growth rate and morphology of the intertidal *Z. noltii* are negatively affected by emergence, and whether its effect is strengthened by the presence of an additional second stressor: shading or high salinity. In addition, we want to test whether the effects of emergence and salinity vary between *Z. noltii* populations from three different origins in western Europe, which are possibly adapted to different local salinity levels. We hypothesize that (1) Emergence has a negative effect on the growth rate and size of the intertidal temperate *Z. noltii* (H1); (2) growth is further reduced when an additional stressor, i.e., shading or high salinity is present (H2), and shading will result in higher aboveground biomass and longer leaves than is commonly found (*De los Santos et al., 2010* (H2b)); and (3) Seagrass origin determines the salinity stress response; plants of populations that grow under high salinity will be less affected in their growth and morphology response than those of low-salinity populations (H3).

## MATERIALS AND METHODS

### Experiment 1

To test the emergence period and shading and their interaction on *Z. noltii*, we examined seagrass growth in a range of four emergence periods (0, 4, 6 and 8 h per 12 h), under control or shaded light conditions. Plants were collected from the Goese Sas tidal flat, Oosterschelde basin, SW Netherlands (51°31.40′N; 3°56.37′E; average salinity of 29.5) (Fig. 1), with permission of the Province of Zeeland (case NB08.068, reference 08033625), transported free floating and stored free-floating in a container with sand-filtered water (salinity of 30, 17 °C, 210 $\mu$E m$^{-2}$ s$^{-1}$ 14 h per day) during three days before the start of the experiment at July 21, 2011. Each experimental planting unit (EPU) consisted of the apical shoot #1, shoot #2, the internode between shoot #1 and #2 and 1 cm of the appending internode between shoot #2 and the cut off shoot #3 (Fig. 2). 80 EPUs were weighed (wet) and separately planted in small trays (20 × 9.5 × 12 cm) filled with an sand/silt mixture (median grain size 169 $\mu$m) from a tidal flat near Bath, Westerschelde estuary, The Netherlands. By having only one EPU per tray, consisting of two shoots, minimal self-shading occurs during emergence. Four trays were in each overflow aquaria (50 × 39 × 26 cm). Four aquaria, each having a different emergence period, were placed in one container; in total five replicate containers were used (so five replicates, four treatments and four pseudo replicates; set-up see Fig. 2) Overflow was created by pumping (300 l/h) sand-filtered Oosterschelde water from the container into the bottom of each aquarium, whereupon water returned into the container. The experimental emergence periods of the seagrass (0, 4, 6, or 8 h per 12 h) were created

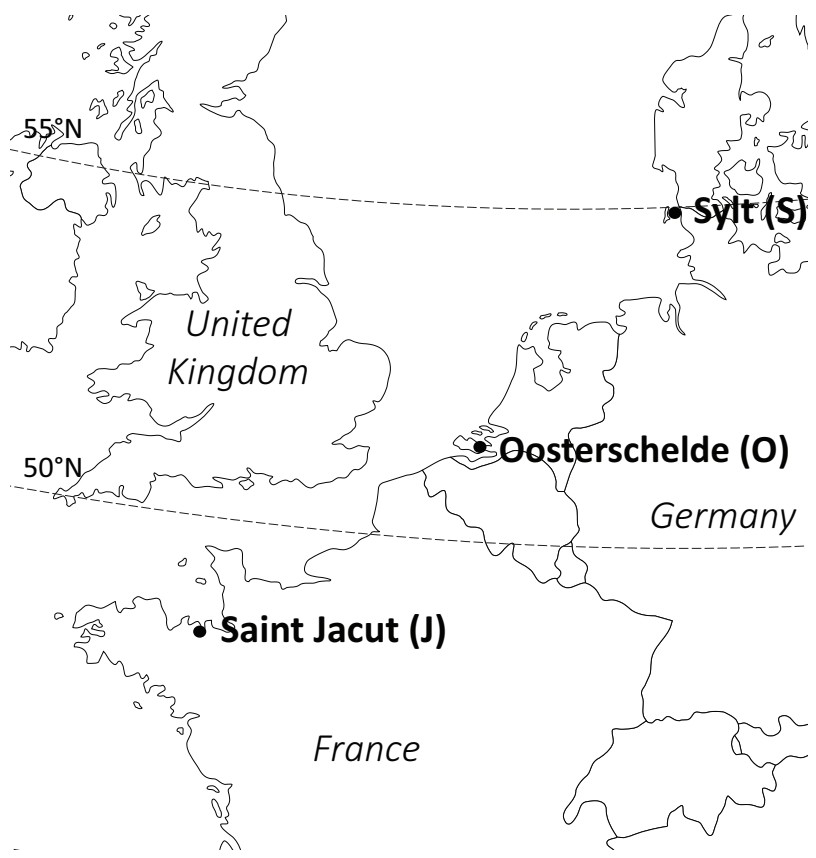

**Figure 1 Origin of the *Zostera noltii* plants used in the experiments (experiment 1: only Oosterschelde, experiment 2: all three populations).**

by automatically switching off the circulation pump, after which the aquaria drained. Emergence periods were chosen for their ecological relevance to seagrass along the natural desiccation gradient (ranging from no desiccation to extreme desiccation): 0 h: subtidal zone (*Z. noltii* grows in this zone in Basque country and Portugal; *Valle et al., 2011* and *Cunha, Assis & Serrao, 2013*, respectively), 4 h: lower intertidal seagrass zone in the Oosterschelde; 6 h: average *Z. noltii* emergence period in the Oosterschelde, 8 h: the upper extreme of *Z. noltii* distribution in the Oosterschelde. Light ($217 \pm 31 \ \mu E \ m^{-2} \ s^{-1}$) was produced by beams of LED-lights emitting a photo spectrum with peaks at 450 (blue) and 670 (red) nm, a spectrum optimized to emit the absorption spectrum for plant photosynthesis. To create low light conditions ($51 \pm 10 \ \mu E \ m^{-2} \ s^{-1}$), black neutral density filters (which reduce the intensity of all wavelengths of light equally, attenuation coefficient of $0.76 \ screen^{-1}$) were placed over half of each aquarium. The photoperiod was set at 14 h of light per day, with one emergence period during the light period and one emergence period was during the dark period. The temperature of the climate controlled room was set at 17 °C. These abiotic settings resemble natural conditions at the beginning of the experiment. Epiphytes (if any) were gently removed weekly. Every week, the entire volume of water of each of the five water containers was refreshed to maintain constant water quality (i.e., salinity, nutrients, etc.)

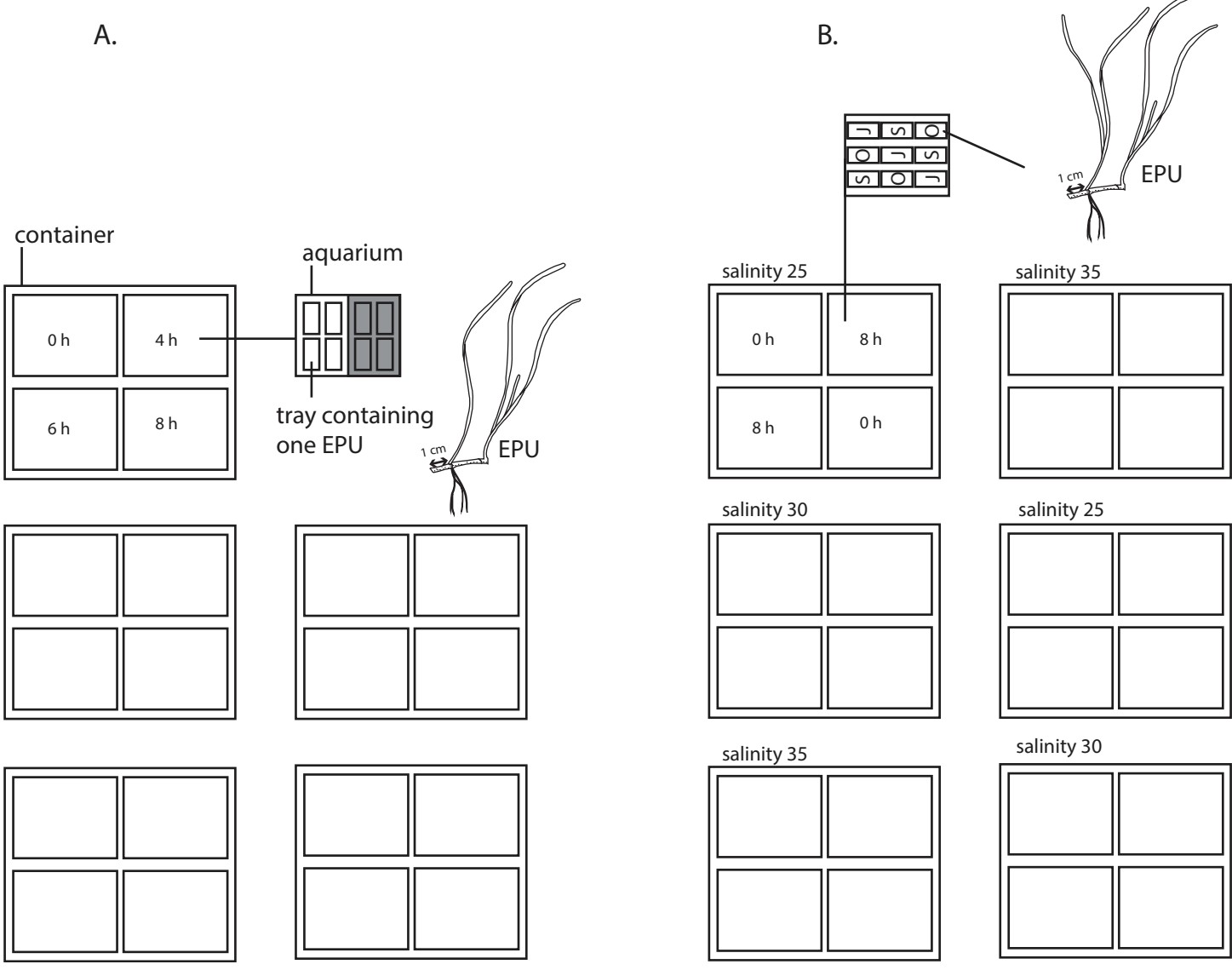

**Figure 2 Experimental set-up of experiment 1 and 2.** (A) Set-up of experiment 1 testing emergence (0, 4, 6 and 8 h per 12 h) and shading (0 and 75% shading of $217 \pm 31$ $\mu$E m$^{-2}$s$^{-1}$, shown as grey shading) in a nested design with five true replicates and four pseudoreplicates. (B) Set-up of experiment two testing three populations of origin (S = Sylt, O = Oosterschelde, J = St Jacut) nested in two replicated emergence regimes (0 and 8 h emergence per 12 h), nested in three salinities (25, 30 and 35). An experimental plant unit (EPU) consists of two *Zostera noltii* shoots (of which one is the apical shoot), plus 1 cm of the adjacent rhizome internode (drawing courtesy of Vanessa González-Ortiz).

among the four replicates within the containers. At the end of the experiment ($t = 92$ days), EPUs were harvested after which their growth response to the treatments was determined by measuring their weight (aboveground and belowground parts separately) and morphology. Relative growth rate (RGR) was based on EPUs wet weight (WW, in grams) and was calculated as: RGR = (ln WW$_{\text{end}}$ − ln WW$_{\text{start}}$)/($\Delta t$), $\Delta t$ is the running time of the experiment in days.

## Experiment 2

To test the effect of emergence, population of origin and salinity and their interaction on *Z. noltii*, we examined seagrass growth of plants originating from three temperate seagrass populations, at two emergence periods, in three natural range salinities. Plants were collected from three locations in Western Europe: Saint-Jacut, (Brittany, France), 48°36′14. 79″N, 2°11′41. 49″W, average salinity of 34.7, Oosterschelde (Southwest Netherlands), 51°53′20.58″N, 3°93′88.35″E, average salinity of 29.5 and Sylt (Wadden Sea, Germany), 54°47′50.77″N, 8°17′43.87″E, average salinity of 29.9 (Fig. 1). At all three locations, water is relatively clear, namely Saint Jacut:, 1.5–2 m secchi depth (2–4 Formazin Nephelometric Units; *Ifremer, 2014*), Oosterschelde: 1.50 m secchi depth (Data Ministry of Infrastructure and Water Management 2002–2009) and Sylt: 1–3.5 m secchi depth (0.5–1.9 m$^{-1}$ light attenuation coefficient at an average tidal range of 1.7 m; *Van Katwijk et al. (1998)* and frequent personal observation second and last author in later years). Plants from Sylt and Saint Jacut were transported to the laboratory within 24 h, stored in wet tissues, kept at 6 °C temp. Plants from Oosterschelde were transported free floating within 3 h. Upon arrival, they were replanted in the laboratory to acclimatize during two months in the same sediments as in experiment 1, in sand-filtered Oosterschelde water (salinity of 31–32, 17 °C, 210 μE m$^{-2}$ s$^{-1}$ 14 h per day) until the start of the experiment on September 30, 2011, (which is at the end of the growing season in the field, but previous studies had shown vigourous growth of this perennial plant during autumn in the laboratory, *Han et al., 2012*). Except for the applied treatments, the experimental set-up resembled that of the first experiment, i.e., same EPUs characteristics, planting trays, overflow aquaria, emergence methods, plant care, artificial lighting, light/dark cycle and climate controlled temperature were used. In this experiment, two containers per salinity treatment were used; in each container, four aquaria were placed with either 0 or 8 h emergence per 12 h, and within each aquarium three pseudoreplicates for each population of origin (Fig. 2). EPUs of each population were weighted wet. To test the response of seagrass to different salinities, overflow aquaria were filled with seawater with a salinity of 25, 30 or 35. Seawater with a salinity of 25 was obtained by mixing demineralized water to the ambient (salinity of 30) seawater, whereas a salinity of 35 was obtained by adding artificial seasalt (Instant Ocean Sea Salt; Instant Ocean, Blacksburg, VA, USA). pH was assumed not to be influenced by the dilution with demineralized water (as was found in a salinity experiment described in *Van Katwijk et al., 1999*; pH data of this experiment are presented in Table S1). To maintain stable salinity levels, salinity was checked at least twice a week and if needed adjusted by adding demineralized water to compensate for evaporation. The water of each container was refreshed after five weeks. Nutrient levels of the water were measured after 28 and 61 days and showed no correlation with salinity treatments (Table S2). EPUs were evenly divided over all treatments ($n = 2$ true replicates × 3 pseudo replicates × 2 emergence replicates = 12 per treatment). At the end of the experiment ($t = 75$ days), EPUs were harvested and measured for weight (aboveground and belowground parts separately) and morphology to determine their growth response. RGR was based on

EPUs wet weight (WW, in grams) and was calculated as: RGR = (ln WW$_{end}$ −
ln WW$_{start}$)/($\Delta t$), $\Delta t$ is the running time of the experiment in days.

## Statistical analysis

The results of experiment 1 were analyzed by linear mixed models with light and
emergence time as fixed factors. As aquaria were nested within containers (Fig. 2) and
EPUs within one aquarium consisted of pseudoreplicates, we included aquarium nested
in container (container/aquarium) as a random factor in our models. Rhizome length,
shoot length, aboveground biomass, belowground biomass, total biomass, RGR and
aboveground/belowground biomass ratio were analyzed by general linear mixed models
(*Pinheiro et al., 2018*) and shoot numbers by a generalized linear mixed model with a
Poisson distribution (*Bates et al., 2018*).

The results of experiment 2 were also analyzed by mixed models with emergence,
salinity and origin as fixed factors. Similar to the set-up of experiment 1, aquaria were
nested in containers (Fig. 2) and we thus included container/aquarium nested as a
random factor in our models. Longest shoot length, rhizome length, aboveground
biomass, belowground biomass, total biomass, RGR, and aboveground/belowground
biomass ratio were analyzed general linear mixed models (*Pinheiro et al., 2018*)
and shoot numbers by a generalized linear mixed model with a poisson distribution
(*Bates et al., 2018*).

Normal distribution of all data was tested on model residuals by means of a Shapiro test
and by looking at the histogram, and data were log- or square root-transformed to
meet model assumptions if necessary.

## RESULTS

Desiccation stress (emergence) significantly reduced aboveground biomass (Fig. 3A,
$P = 0.010$) and RGR (Fig. 3F, $P = 0.045$) (Table 1); aboveground biomass was reduced
by 27% and RGR by 32% in the 8 h emergence time treatment compared to the
completely submerged treatment (0 h emergence time). In line with this, emergence
also significantly reduced rhizome length (Fig. 3G, $P = 0.049$) in the 6 and 8 h
emergence treatment by 26–28%. Next to desiccation stress, Shading (light reduction)
also significantly reduced the RGR (Fig 3F, $P = 0.039$) by 18%. In addition, shading
significantly reduced belowground biomass by 33% (Fig. 3B, $P < 0.001$) and total biomass
by 23% ($P = 0.010$), thereby strongly increasing the aboveground/belowground
biomass ratio (Fig. 3D, $P < 0.001$). Plant morphology also changed significantly under
light limitation stress; leaves grew 21% longer when shaded (Fig. 3H, $P = 0.011$).
Although we found clear effects of desiccation and shading stress on growth
(RGR, biomass) and morphology (leaf length and rhizome length), we did not observe
any interactive effects of desiccation and light limitation stress.

The second experiment included exposing *Z. noltii* from three different locations in
Western Europe (France, the Netherlands and Germany) to desiccation stress (0 vs. 8 h
emergence) and three different salinities (25, 30, 35 ppt). Similar to the first
experiment, we found that desiccation stress (8 h emergence) reduced RGR

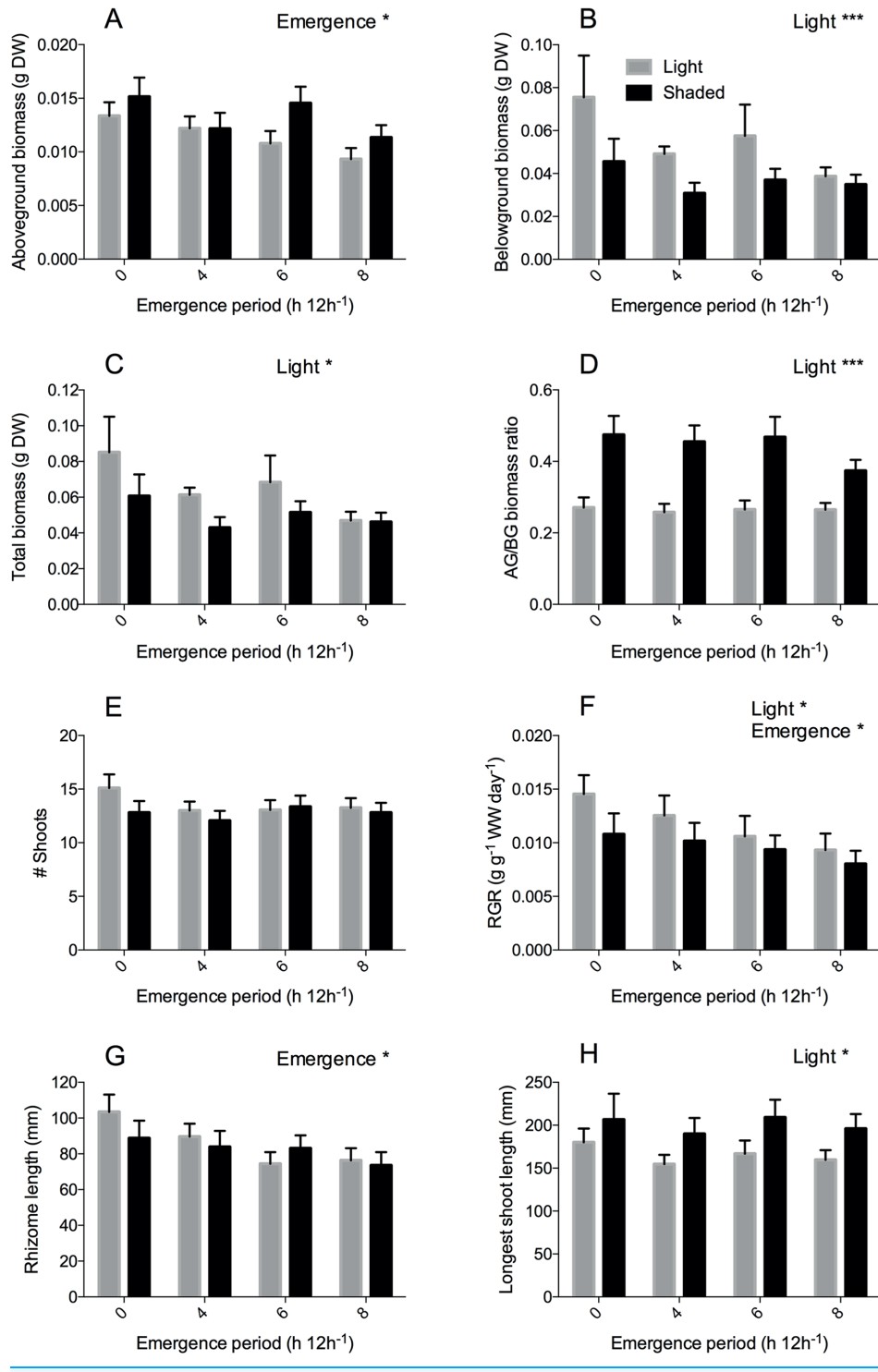

**Figure 3** *Zostera noltii* **development after 60 days in relation to emergence time (x-axis) under light (grey bars) and shaded (black bars) conditions.** (A) Aboveground biomass, (B) belowground biomass, (C) Total biomass, (D) aboveground/belowground (AG/BG) biomass ratio, (E) shoot numbers (# shoots), (F) relative growth rate (RGR) of wet weight (WW), (G) rhizome length and (H) longest shoot length. Statistical results are displayed in the upper right corner of each panel, $0.01 < P < 0.05 = *$, $P < 0.001 = ***$. Detailed statistical results are displayed in Table 1. Error bars represent standard errors (SEM).

| Table 1 Effects of light and emergence on *Zostera noltii*. | | | | | | | |
|---|---|---|---|---|---|---|---|
| **Factor** | **Test** | **Transformation** | **Treatment** | **F value** | **DF** | **P** | |
| DW AG | lme | log | Light | 3.531 | 1 | 0.060 | n.s. |
| | | | **Emergence** | **11.290** | **1** | **0.010** | * |
| | | | Light × emergence | 3.965 | 1 | 0.265 | n.s. |
| DW BG | lme | log | **Light** | **14.728** | **1** | **<0.001** | *** |
| | | | emergence | 3.022 | 1 | 0.388 | n.s. |
| | | | light × emergence | 3.597 | 1 | 0.308 | n.s. |
| DW total | lme | log | **Light** | **6.569** | **1** | **0.010** | * |
| | | | Emergence | 3.667 | 1 | 0.300 | n.s. |
| | | | Light × emergence | 3.498 | 1 | 0.321 | n.s. |
| AG/BG ratio | lme | sqrt | **Light** | **52.640** | **1** | **<0.001** | *** |
| | | | Emergence | 1.560 | 1 | 0.669 | n.s. |
| | | | Light × emergence | 2.404 | 1 | 0.493 | n.s. |
| # shoots | glmer | – | Light | 2.185 | 1 | 0.139 | n.s. |
| | | | Emergence | 3.165 | 1 | 0.367 | n.s. |
| | | | Light × emergence | 2.545 | 1 | 0.467 | n.s. |
| RGR WW | lme | None | **Light** | **4.240** | **1** | **0.039** | * |
| | | | **Emergence** | **8.042** | **1** | **0.045** | * |
| | | | Light × emergence | 0.955 | 1 | 0.812 | n.s. |
| Rhizome length | lme | None | Light | 0.480 | 1 | 0.489 | n.s. |
| | | | **Emergence** | **7.865** | **1** | **0.049** | * |
| | | | Light × emergence | 2.169 | 1 | 0.538 | n.s. |
| Longest shoot | lme | log | **Light** | **6.421** | **1** | **0.011** | * |
| | | | Emergence | 1.388 | 1 | 0.708 | n.s. |
| | | | Light × emergence | 0.364 | 1 | 0.947 | n.s. |

**Notes:**
Statistical results of Experiment 1. Non-significant test results are marked with "ns," whereas significant test results are bold and marked with stars: $0.01 < P < 0.05 =$*, $P < 0.001 =$***.
lme, general linear mixed model; glmer, generalized linear mixed model with a Poisson distribution; DF, degrees of freedom; P, P-value; DW, dry weight; WW, wet weight; AG, above ground biomass; BG, below ground biomass; RGR WW, Relative growth rate; #, number; SQRT, square root; log, logarithm.

(Fig. 4F, $P < 0.001$) and aboveground biomass (Fig. 4A, $P = 0.003$) by 25% and 19% respectively (Table 2). In addition, belowground biomass (Fig. 4B, $P < 0.001$) and total biomass ($P = 0.001$) were also reduced by desiccation stress by 32% and 29% (Table 2). Location of origin affected morphology and biomass of the plants; rhizomes (Fig. 4G, $P < 0.001$) and leaves (Fig. 4H, $P < 0.001$) were smaller and aboveground (Fig. 4A, $P < 0.001$), belowground (Fig. 4B, $P < 0.001$) and total biomass (Fig 4C, $P < 0.001$) lower in a gradient from France (Saint Jacut) to Germany (Sylt) (Table 2). Leaves and rhizomes from the French *Z. noltii* were 22% longer than the German plants and above- and belowground biomass was approximately 2.4 times larger as compared to Sylt plants at the end of the experiment (pooled results, Fig. 4). As RGR did not differ between the origins of the populations (e.g., RGR of Saint Jacut was 1.06 times the RGR of Sylt), the size differences at the end of the experiment reflect the differences at the beginning of the experiment. Similar plant size differences between these populations are observed in other years or

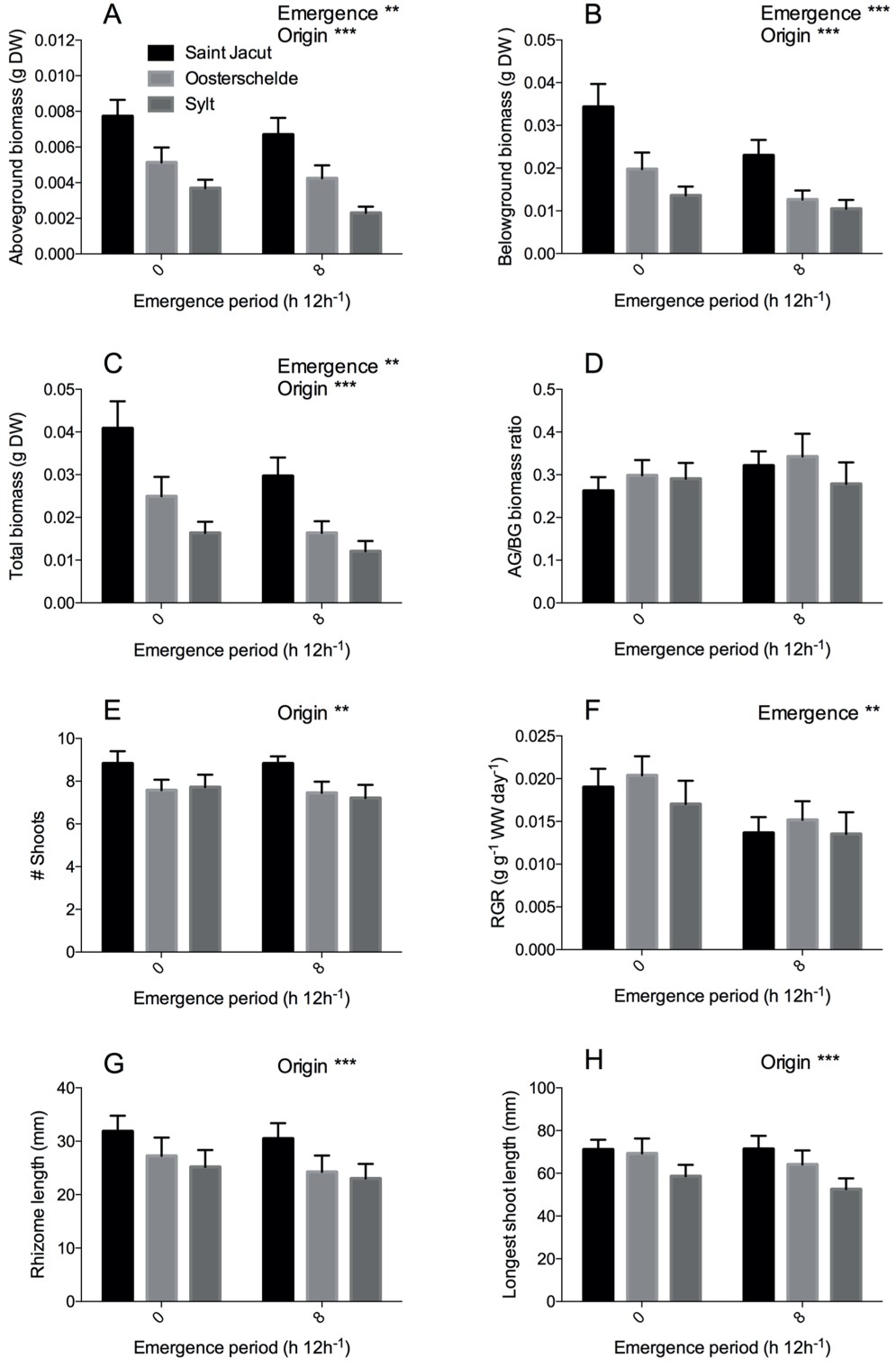

**Figure 4 *Zostera noltii* growth response after 75 days of combined salinity and emergence treatment.** Salinity treatments are pooled as no significant differences were found between treatments of all measured parameters (Table 2). Locations of origin are displayed in the following colors: Saint Jacut in black, Oosterschelde in light grey and Sylt in dark grey. (A) Aboveground biomass, (B) belowground biomass, (C) total biomass, (D) aboveground/belowground (AG/BG) biomass ratio, (E) shoot numbers (# shoots), (F) relative growth rate (RGR) of wet weight (WW), (G) rhizome length and (H) longest shoot length. Statistical results are displayed in the upper right corner of each panel, $0.001 < P < 0.01 = **$, $P < 0.001 = ***$. Error bars represent standard errors (SEM).

seasons (*Soissons et al., 2018*; *Govers, 2014*). Plants originating from the Netherlands (Oosterschelde) were in the middle for all measured traits. In addition, total shoot numbers were also significantly higher (Fig. 4E, $P = 0.004$; Table 2) in units with plants from France than in units with Dutch and German plants. Despite these differences in morphology and biomass, seagrass plants from different origins were similarly affected by emergence and salinity as we found no interactive effects for any of our parameters (salinity × origin × emergence). Surprisingly, we found no effect of our salinity treatment on any growth or morphological parameters, indicating a broad salt tolerance of *Z. noltii* (Table 2).

## DISCUSSION

Emergence had a negative effect on seagrass growth in both experiments confirming our hypothesis (H1). Increasing duration of air exposure hampered seagrass growth, reflected by a smaller RGR and reduced aboveground biomass of the plants as compared to the control group that was never subjected to emergence. The reduced growth might be the consequence of loss of photosynthetic capacity due to desiccation damage to the leaves (*Leuschner, Landwehr & Mehlig, 1998*; *Vermaat, Verhagen & Lindenburg, 2000*; *Fernandez-Torquemada & Sanchez-Lizaso, 2005*; *Shafer, Sherman & Wyllie-Echeverria, 2007*). In the field, the negative effect of desiccation was demonstrated by the higher biomass found in depressions (which retain water during low tide) as compared to elevations (dry at low tide) within the same bed (*Van Tussenbroek et al., 2016*). Concurrently, these authors found lower sexual reproductive efforts in the depressions, in line with the general notion that reproductive efforts increases with increasing stress in seagrasses (*Cabaco & Santos, 2012*). Further growth reduction from desiccation can be expected in situ as sediment trapping by *Z. noltii* often causes the plant to grow on elevations, making them more prone to desiccation (*Reise & Kohlus, 2008*; *Van der Heide et al., 2010*; *Van Tussenbroek et al., 2016*). Furthermore, desiccation damaged leaves may be easier torn by waves than undamaged leaves, leaving significantly shorter leaves for photosynthesis (*Vermaat et al., 1993*; *Boese et al., 2003*), and reducing water retention by the leaves (*Fox, 1996*), which is density dependent (*De Fouw et al., 2016*).

Shading resulted in longer leaves, but reduced belowground biomass and RGR. Aboveground/belowground biomass ratio increased (experiment 1), similar to previous studies without emergence treatments (*Vermaat et al., 1993*; *Peralta et al., 2002*; *Cabaco, Machas & Santos, 2009*). Plants apparently invested more in aboveground

**Table 2 Results of statistical tests testing the main and combined effects of emergence time, salinity and population origin (location) on seagrass growth and morphology.**

| Factor | Test | Transformation | Treatment | DF | *F* value | *P* | |
|---|---|---|---|---|---|---|---|
| DW AG | lme | sqrt | **Emergence** | **1** | **12.571** | **0.003** | ** |
| | | | Salinity | 1 | 4.109 | 0.138 | n.s. |
| | | | **Origin** | **2** | **50.517** | **<0.001** | *** |
| | | | Emergence × salinity | 2 | 1.201 | 0.328 | n.s. |
| | | | Emergence × origin | 2 | 0.865 | 0.423 | n.s. |
| | | | Salinity × origin | 4 | 0.358 | 0.839 | n.s. |
| | | | Emergence × salinity × origin | 4 | 2.042 | 0.091 | n.s |
| DW BG | lme | sqrt | **Emergence** | **1** | **20.036** | **<0.001** | *** |
| | | | Salinity | 1 | 0.801 | 0.7871 | n.s. |
| | | | **Origin** | **2** | **37.318** | **<0.001** | *** |
| | | | Emergence × salinity | 2 | 1.224 | 0.322 | n.s. |
| | | | Emergence × origin | 2 | 0.710 | 0.493 | n.s. |
| | | | Salinity × origin | 4 | 0.664 | 0.617 | n.s. |
| | | | Emergence × salinity × origin | 4 | 0.591 | 0.670 | n.s |
| DW total | lme | sqrt | **Emergence** | 1 | 15.312 | 0.001 | ** |
| | | | Salinity | 1 | 0.180 | 0.843 | n.s. |
| | | | **Origin** | **2** | **39.652** | **<0.001** | *** |
| | | | Emergence × salinity | 2 | 1.150 | 0.343 | n.s. |
| | | | Emergence × origin | 2 | 0.182 | 0.834 | n.s. |
| | | | Salinity × origin | 4 | 0.690 | 0.600 | n.s. |
| | | | Emergence × salinity × origin | 4 | 1.393 | 0.238 | n.s |
| AG/BG ratio | lme | sqrt | Emergence | 1 | 1.151 | 0.300 | n.s. |
| | | | Salinity | 1 | 0.460 | 0.670 | n.s. |
| | | | Origin | 2 | 0.919 | 0.401 | n.s. |
| | | | Emergence × salinity | 2 | 0.726 | 0.500 | n.s. |
| | | | Emergence × origin | 2 | 1.577 | 0.210 | n.s. |
| | | | Salinity × origin | 4 | 0.344 | 0.848 | n.s. |
| | | | Emergence × salinity × origin | 4 | 1.355 | 0.252 | n.s |
| # shoots | glmer | n.a. | Emergence | 1 | 0.285[1] | 0.593 | n.s. |
| | | | Salinity | 1 | 1.491 | 0.222 | n.s. |
| | | | **Origin** | **2** | **10.793** | **0.004** | ** |
| | | | Emergence × salinity | 1 | 0.013 | 0.910 | n.s. |
| | | | Emergence × origin | 2 | 0.341 | 0.843 | n.s. |
| | | | Salinity × origin | 2 | 1.459 | 0.482 | n.s. |
| | | | Emergence × salinity × origin | 2 | 0.162 | 0.922 | n.s |
| RGR WW | lme | sqrt | **Emergence** | **1** | **17.810** | **<0.001** | *** |
| | | | Salinity | 1 | 1.030 | 0.457 | n.s. |
| | | | Origin | 2 | 1.770 | 0.174 | n.s. |
| | | | Emergence × salinity | 2 | 0.690 | 0.518 | n.s. |
| | | | Emergence × origin | 2 | 0.290 | 0.746 | n.s. |
| | | | Salinity × origin | 4 | 0.730 | 0.572 | n.s. |
| | | | Emergence × salinity × origin | 4 | 2.230 | 0.068 | n.s. |

*(Continued)*

**Table 2** (*continued*).

| Factor | Test | Transformation | Treatment | DF | *F* value | *P* | |
|---|---|---|---|---|---|---|---|
| Rhizome length | lme | none | Emergence | 1 | 2.033 | 0.156 | n.s. |
| | | | Salinity | 1 | 2.948 | 0.161 | n.s. |
| | | | **Origin** | **2** | **8.581** | **<0.001** | *** |
| | | | Emergence × salinity | 1 | 0.481 | 0.489 | n.s. |
| | | | Emergence × origin | 2 | 0.101 | 0.903 | n.s. |
| | | | Salinity × origin | 2 | 0.250 | 0.779 | n.s. |
| | | | Emergence × salinity × origin | 2 | 0.800 | 0.451 | n.s. |
| Longest shoot | lme | none | Emergence | 1 | 1.189 | 0.277 | n.s. |
| | | | Salinity | 1 | 1.721 | 0.260 | n.s. |
| | | | **Origin** | **2** | **9.918** | **<0.001** | *** |
| | | | Emergence × salinity | 1 | 0.543 | 0.462 | n.s. |
| | | | Emergence × origin | 2 | 0.450 | 0.638 | n.s. |
| | | | Salinity × origin | 2 | 0.332 | 0.718 | n.s. |
| | | | Emergence × salinity × origin | 2 | 0.670 | 0.513 | n.s. |

Notes:

Non-significant test results are marked with "ns," whereas significant test results are in bold, marked with *** for $P < 0.001$ and **$0.001 < P < 0.001$.

DF, degrees of freedom; *P*, *P*-value; DW, dry weight; WW, wet weight; AG, above ground biomass; BG, below ground biomass; RGR WW, Relative growth rate; #, number; sqrt, square root; log, logarithm; lme, general linear mixed model; glmer, generalized linear mixed model.

[1] Indicates chi-square values of the generalized linear mixed model rather than *F* values.

biomass than in belowground biomass. Confirming our hypothesis (H2), shading added to the negative effect of emergence on RGR. Effects were additive (no interactive effect, so no antagonistic or synergistic effects). The combination is thus more stressful for the plants than singular effects and may explain the strong reduction in net photosynthesis in *Z. noltii* beds during low tide as compared to high tide, due to self-shading by the leaves lying on top of each other during low tide (*Clavier et al., 2011*; *Ouisse, Migné & Davoult, 2011*). When shading in the intertidal is caused by algal overgrowth, desiccation of the plants during low tide may be less severe as the leaves are kept wet by the algal cover. However, in such cases, suffocation, sulfide and ammonium toxicity pose a threat to seagrasses (*Goodman, Moore & Dennison, 1995*; *Den Hartog, 1996*; *Holmer & Nielsen, 2007*; *Govers et al., 2014*). In addition, a stronger reduction in net growth may be expected in the field as the resulting longer leaves experience more wave induced drag force than the shorter leaves that develop under ambient light conditions (*Bouma et al., 2005*; *La Nafie et al., 2012*).

The tested salinity range (salinities of 25, 30 and 35, experiment 2) did not affect plant growth over the course of the experiment, which contrasts our hypothesis (H3). Apparently, *Z. noltii* was not stressed by salinities of 30 and 35, probably due to its ability to acclimatize to salinity changes (*Touchette, 2007*), not resulting in reduced growth (production of biomass) as observed for *Z. marina* by *Kamermans, Hemminga & de Jong (1999)*. This is supported by presence of dense *Z. noltii* beds in Banc d'Arguin, Mauritania with salinity levels of over 40 (*Vermaat et al., 1993*), whereas *Z. marina* has an optimum at salinities as low as salinity 25 (*Nejrup, Brammer & Pedersen, 2008*).

Whereas hypothesis H2 (additional stressors aggravate the effects of emergence) was confirmed with regards to shading, it was not confirmed regarding salinity in the range tested. Considering that salinity did not have an effect at all within the tested range, this is not surprising. Perhaps this implies that *Z. noltii* is more of a generalist than a specialist in terms of the environmental extremes it can withstand, making it more of an opportunistic pioneer species instead of a climax community species, as compared to *Z. marina*.

The effects of salinity and desiccation stress (experiment 2) did not differ between plants of different origin. This contrasts our expectation that plants that are used to relatively low salinities would encounter more osmotic stress than those that are already used to relative high salinity conditions, as was shown for *Z. noltii* comparing a Spanish and Dutch population in salinity 15 and 35 (*Vermaat, Verhagen & Lindenburg, 2000*) and for *Z. marina* in a narrower or lower salinity range (salinity 22–29, *Van Katwijk et al., 1999*; salinity 2–25, *Salo, Pedersen & Bostrom, 2014*). Thus, *Z. noltii* plants that normally grow at a salinity of 35 grew equally well at salinities of 25 or 30, and plants used to salinities of 29–30 grew equally well at salinities of 25 or 35. Perhaps this flexibility may be explained by plants originating from estuarine and shallow coastal conditions (as we used in our experiments) being more used to frequent variations in osmotic stress and thus better able to make fast physiological adaptations, compared to plants from more osmotically stable, true marine environments. Although plants from the three populations differed in plant size and morphology (from big (Saint Jacut), medium (Oosterschelde) to small (Sylt)), in line with the trend over a broader latitudinal gradient for this species (South Spain to Sylt; *Soissons et al., 2018*), their plant size did not influence their ability to cope with emergence or salinity stress.

## CONCLUSIONS AND ECOLOGICAL IMPLICATIONS

In this study, we found that desiccation stress imposed by emergence and shading have a negative effect on *Z. noltii*. When combined, they show additive effects, there are no synergistic or antagonistic effects. This makes plants vulnerable to such combination (although synergistic effects would enhance this vulnerability even more). The three populations tested show distinctive size difference, but did not respond differentially to emergence and salinity. This indicates that the populations show morphotypic rather then ecotypic variation regarding emergence and salinity within the range tested.

Desiccation is likely to determine the upper distribution of *Z. noltii*, although other factors than physiological factors (such as predation, competition for space and resources) can also be important under field conditions. The question arises: why don't plants grow towards the mean low water level in our research areas, where emergence times are shorter? Our study shows that shorter and absent emergence periods are favorable for *Z. noltii*. Although seagrass meadows are often light limited at increasing depth (*Philippart, 1995*; *Ralph et al., 2007*; *Cabaco, Machas & Santos, 2009*; *Van der Heide et al., 2010*), light is not likely a limiting factor in the habitats of the three tested populations (see Materials and Methods). It is probable that, water and/or sediment dynamics may explain the absence of *Z. noltii* in the lower ranges of the intertidal

(*Suykerbuyk et al., 2016a*, *2016b*). Still, in more southern ranges of its distribution, *Z. noltii* occupies the whole intertidal range and expands even in the higher subtidal (e.g., Basque country, *Valle et al., 2011*; Portugal, *Cunha, Assis & Serrao, 2013*) and submerged in seas where tides are near absent (Mediterranean, *Green & Short, 2003*). Further research is required to assess key factors in determining the depth limit of mid-intertidal *Z. noltii* beds. However, from our study it is clear that (i) emersion period is a factor controlling the upper limit, (ii) shading (for example by self-shading during emersion) likely lowers the upper limit, as the effects shading and emergence were additive, and (iii) this upper limit is not affected by salinity or origin of the population. For restoration purposes, there is no need to carefully select donor populations regarding salinity regime within the range tested, as the plants of different origins (salinity 30 and 35) were not influenced by applied salinities (25, 30, 35).

## ACKNOWLEDGEMENTS

The authors are grateful to Jan Vermaat for his expert opinion on the initial research plans. We would also like to acknowledge Ragnhild Asmus and Dominik Kneer (Sylt) and Laurent Godet and Jérôme Fournier (Saint Jacut) for their cooperation and help in the collection of *Z. noltii*. Lastly, we thank Vanessa González-Ortiz for providing the artwork of the experimental plant unit.

### Funding
This work was supported by Projectbureau Zeeweringen, Middelburg, the Netherlands. The funders had no role in study design, data collection and analysis, decision to publish, or preparation of the manuscript.

### Grant Disclosures
The following grant information was disclosed by the authors:
Projectbureau Zeeweringen.

### Competing Interests
The authors declare that they have no competing interests.

### Author Contributions

- Wouter Suykerbuyk conceived and designed the experiments, performed the experiments, analyzed the data, prepared figures and/or tables, authored or reviewed drafts of the paper, approved the final draft.
- Laura L. Govers conceived and designed the experiments, performed the experiments, analyzed the data, prepared figures and/or tables, authored or reviewed drafts of the paper, approved the final draft.
- W.G. van Oven conceived and designed the experiments, performed the experiments, analyzed the data, prepared figures and/or tables, authored or reviewed drafts of the paper, approved the final draft.

- Kris Giesen performed the experiments, analyzed the data, authored or reviewed drafts of the paper, approved the final draft.
- Wim B.J.T. Giesen conceived and designed the experiments, authored or reviewed drafts of the paper, approved the final draft.
- Dick J. de Jong conceived and designed the experiments, authored or reviewed drafts of the paper, approved the final draft.
- Tjeerd J. Bouma conceived and designed the experiments, authored or reviewed drafts of the paper, approved the final draft.
- Marieke M. van Katwijk conceived and designed the experiments, prepared figures and/or tables, authored or reviewed drafts of the paper, approved the final draft.

## Field Study Permissions

The following information was supplied relating to field study approvals (i.e., approving body and any reference numbers):

The Province of Zeeland gave permission to perform research in the study area to NIOO/CEME (case NB08.068, reference 08033625), the institute that is now named 'NIOZ,' where several authors are/were employed.

## Data Availability

The raw data are provided in a Supplemental File.

## Supplemental Information

Supplemental information for this article can be found online at http://dx.doi.org/10.7717/peerj.5234#supplemental-information.

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
