# Peer review of "Living in the intertidal: desiccation and shading reduce seagrass growth, but high salinity or population of origin have no additional effect"

_PeerJ, doi:10.7717/peerj.5234_

## Round 0.1 · original submission · Major Revisions

Both reviewers identified the same major issue with the manuscript: a lack of clarity in the description of the experimental design and the statistical analyses, which made it difficult to judge whether the experiment and statistics are of sufficient quality. The quality of the experimental design and statistical analyses are the most important criteria for review at PeerJ, so the authors should pay special attention to the reviewers' comments on these topics. In instances where treatments are pseudo-replicated (as suspected by reviewer 1), the authors should acknowledge this reality and describe the limitations on their conclusions according to this weakness. I agree with reviewer 2's request that the authors should more fully explain the utility (or lack thereof) of testing for a synergistic effect of low light and dessication, and also more appropriately constrain their conclusions regarding this interaction based on the light and dessication conditions typically experienced by 'natural' seagrass populations.

Both reviewers also pointed out issues related to language (e.g., missing words or phrases in some places; potential contradictions between their results and their statements) and formatting (e.g., the tables which have words and numbers that should be shown on one row only, but are spilling into the next row). The authors should carefully check the entire manuscript for clarity and accuracy of the writing and for appropriate formatting.

Reviewer 1 ·

Basic reporting

The language of the manuscript requires a lot of work. I am wondering if the authors accidentally submitted a wrong version of the manuscript, as in some places it is obvious that text is missing. For example L213-215 and L247-248. Some of the abbreviations were not introduced (L194-195, Table 2), some abbreviations were not used consistently (e.g. aboveground and belowground were sometimes abbreviated as AG and BG, and other times A and B and in other places written out). Also some of the results described in the text or statistical tables do not match with the information in the figures. For example, the shoots could not have been 121% longer (L222) as based on the Fig 2F, these were ca 20-30% longer. Also if the figures really are mean ±SEM, I have hard time believing the statistical analyses. As for example the error bars for in Fig.2C overlap consistently, and the 95% CI should be much larger that SEM, I cannot understand how the authors managed to find these tiny differences statistically significant.

Experimental design

My biggest concern is the experimental design used in both experiments. In its current form, it seems that the authors only had one replicate per treatment and treatment combination. It seems that they had all the ”replicate” plants in the same unit/aquaria for the combined treatment effects. This means that they have used individual plant units (2 connected ramets) as replicate in the statistical assessment of the results. However, these units are merely pseudoreplicates, and they should have used several experimental units with plants to replicate for each treatment and treatment combination. In the second experiments, the design is described as nested split-plot design, with population as partly nested (split-plot) under emergence, which then should be nested under salinity. However, based on the description of the statistical testing, this is not how it was analyzed. Additionally, only having one tray of plants per treatment combination, testing the results is not even possible. Not even with the potential hidden replication due to use of several salinities. However, as the material and methods section was not very clearly written, I am not completely sure I understood it correctly. With this complicated design, I suggest the authors add an additional figure illustrating the desing for both experiments.

Validity of the findings

See point 2. Due to the most likely faulty statistical analyses, I cannot assess the validity of the findings.
However, if the statistical methods would be ok, the Discussion section would need clearly more work. Now the Discussion is very superficial and instead of discussing the results in a larger context, mainly turns the discussion to what was not tested in the experiments. The authors end up drawing their main conclusion about something they did not test at all in this experiment.
There are interesting findings in the paper, and I believe the ms would be much more interesting if the authors concentrated more on these. (See general comments below)

Additional comments

Reviewer comments for manuscript ”Living in the intertidal; desiccation and shading reduce seagrass growth, but high salinity or population of origin have no additional effect” (#22711) by Suykerbuyk et al.

The manuscript reports findings from two experiments testing how desiccation together with other potentially affecting factors (shading, salinity and plant population) affect the growth and morphology of the seagrass Zostera noltii. While the findings, and especially the lack of interaction of different tested factors, are interesting, I have several concerns about the manuscript, and cannot recommend it to be acceptedwithout further knowledge on the experimental design. Currently, the material and methods section is lacking some important information about the methods and design used. Even if the design turns out to be ok, the manuscript requires soem major revisions.

Firstly, my biggest concern is the experimental design used in both experiments. In its current form, it seems that the authors only had one replicate per treatment and treatment combination. It seems that they had all the ”replicate” plants in the same unit/aquaria for the combined treatment effects. This means that they have used individual plant units (2 connected ramets) as replicate in the statistical assessment of the results. However, these units are merely pseudoreplicates, and they should have used several experimental units with plants to replicate for each treatment and treatment combination. In the second experiments, the design is described as nested split-plot design, with population as partly nested (split-plot) under emergence, which then should be nested under salinity. However, based on the description of the statistical testing, this is not how it was analyzed. Additionally, only having one tray of plants per treatment combination, testing the results is not even possible. Not even with the potential hidden replication due to use of several salinities. However, as the material and methods section was not very clearly written, I am not completely sure I understood it correctly. With this complicated design, I suggest the authors add an additional figure illustrating the desing for both experiments.

Secondly, the language of the manuscript requires a lot of work. I am wondering if the authors accidentally submitted a wrong version of the manuscript, as in some places it is obvious that text is missing. For example L213-215 and L247-248. Some of the abbreviations were not introduced (L194-195, Table 2), some abbreviations were not used consistently (e.g. aboveground and belowground were sometimes abbreviated as AG and BG, and other times A and B and in other places written out). Also some of the results described in the text or statistical tables do not match with the information in the figures. For example, the shoots could not have been 121% longer (L222) as based on the Fig 2F, these were ca 20-30% longer. Also if the figures really are mean ±SEM, I have hard time believing the statistical analyses. As for example the error bars for in Fig.2C overlap consistently, and the 95% CI should be much larger that SEM, I cannot understand how the authors managed to find these tiny differences statistically significant.

The authors describe that they did model selection for the glms and lme’s and removed any non-significant main factor. In table 1 and 2 they, however, include all the factors. It is unclear if these are from the original model, or if they combined these for the different factors.
Further, the main findings according to the authors is that shading reduced plant growth, but increased aboveground biomass. It is clear from the data that this is due to increaed investment in aboveground biomass compared to belowground biomass. This reduced belowground investment and shoft to aboveground production is not considered at all, which in my opinion is one of the most interesting findings in the manuscript.

Some more minor comments:
Synergistic is not the synonym for interactive effects. Stress effects can be either interactive (e.g. synergistic or antagonistic) or additive. The terminology should be used consistently throughout the manuscript.
L28-29. PSU is no longer used for salinity.
L46-49. The authors did not test sediment properties, so haw can their main conclusions be about that?
L81. Also when desiccated or only submerged?
L106-109. How were the day length effects? These should also be reported if they are mentioned in the text
L111. Additive effects instead of ”they add-up”. Same throughout the manuscript
L115. What is referred to by ”these effects”
L118. What is referred to by ”They”?
L118-119. These information belong to M&M, not introduction
L119-123. No morphology-related hypoteses? Compare statement on L113-115.
L130. How were these stored? Transplanted in sediment, or kept freefloting in tanks? Photoperiod? And how were they transported to the location? Same question goes for the experiment 2 plants.
L135. Does using untreated sediment mean that there were infauna in the experiments?
L151. How was the photoperiod in relation to desiccation?
L160. How was the initial WW quantified? For each specific plant?
L170-171. Reference?
L186. N=12 per population or in total?
L193 & 199. An experiment is not analyzed, the results are
L199. If the authors want to test general population effects, then population should be random factor, not fixed.
L213-215. Text missing
L231-232. These results are not described for desiccation anywhere
L236. Is 22% referring to the initial plant size differences? I suggest considering using relative data for size and bm for different populations, to easier see differences in plant responses, instead of their different original and final size. Also, the initial differences should be given in material and methods section.
L244-245. These results are not shown anywhere
L271-276. Not really that relevant to this study. Teh authors should not try to simply discuss what differs between lab and field, but to try to discuss how and why they obtained results like they did and how these could reflect in more natural conditions.
L285-286. Z. Marina results here not relevant
L297-298. How is salinity of 2-25 narrower than salinity of 15-35?
L323-325. I suggest that if the authors want to bring discussion to the differences between sites (Should be introduced in M&M, instead of Discussion), they should refer to newer literature. Water clarity can certainly change a lot during 24 years.
L335. Personal observations of the last author?

Reviewer 2 ·

Basic reporting

The manuscript by Suykerbuyk and co-workers aims to assess the effects of desiccation and light limitation in affecting growth and morphology of intertidal populations of the seagrass Z. noltii. Moreover, authors seek to understand the differential effects of salinity range on populations pre-adapted to different salinities.
In general, I found that the manuscript has some problems that impair at the moment the publication in a high rank journal such as PeerJ.
First, it is not clear why authors tested the possible synergic effect of low light and desiccation. When exposed, plants experience higher light that can really represent a further stressor. Low light can negatively affect plant growth, but it does not occur at the same moment when desiccation occurs.
Second, the experimental setup is not well explained and makes it difficult to evaluate results. A figure clearly detailing the experimental design should be presented for the two experiments.
Third, the interpretation of some results is not clear.

Experimental design

In order to achieve their goals, authors perform two distinct experiments in a closed aquarium system and measure growth and morphological traits after 2-3 months of treatment. The experimental design is not well explained, which also impairs a correct evaluation of the appropriateness of the statistics utilized. For example, in exp 1 it is said that 20 EPUs were utilized per treatment and each EPU was planted individually in a separate tray. Hence, they were placed in aquaria (how many EPU?). Are the aquaria representing the treatment? It is to say 20 EPUs per aquarium? Or are the aquaria representing the n=4 replicates and 5 EPUs were placed in each of four aquaria used for each treatment? Since half of each aquarium was shaded by filters, how many were shaded and how many represented the control? If the number of EPUs per aquaria is 5, the treated and the control ones cannot be of the same number.
Maybe 20 EPUs were planted in each of four aquaria utilized for each treatment. In this case, total number of EPUs per treatment should be 80 and not 20.

In exp. 2, the experimental setup is apparently clarified in the raw data provided with the submission. Nevertheless, the scheme does not correspond to the possible setup explained in the text. At line 174-175 it is stated that: “In this experiment, 2 containers per salinity treatment were used, in each container 2 aquaria were placed, and within each aquarium 3 pseudoreplicates for each population of origin.” In the raw data 6 containers are represented, with 4 aquaria for each container.
Moreover, here it should be explained how the acclimation period was performed. Were plants acclimated to the same salinity conditions? For how long? Length of the acclimation phase is very important in common garden experiments.

The species is well known to tolerate high salinity ranges and the selected populations grow in a salinity range that is within the tolerance limits of the species (two populations naturally experience average salinity of 29.5-29.9, while one grows at a salinity of 34.7). How were the experimental salinity levels decided?

Since the nested experimental design is not clear, it is not possible to evaluate the appropriateness of the statistics (the fixed and random factors, etc…) into detail.

Validity of the findings

Results are well presented and figures are all correct and necessary. In my version, tables are not well formatted.
Main conclusion of experiment 1 is that both light reduction and desiccation impact seagrass growth, with light affecting also the BG/AG ratio and leaf length. Plants are not able to invest in the below ground tissue when light is limiting, and invest more in the photosynthetic tissue. The two stressors do not show significant interaction.
In the second experiment salinity does not have any effect on shoots growth and morphology and only desiccation produced significant results. Plants form the Northernmost population, the one with higher salinity, were bigger and this reflects all along the experiment.
Line 213; there is something wrong at the beginning of the sentence
Line 232: delete: “Z. noltii origin”

At the beginning of the section, authors discuss the effect of emergence, desiccation, in comparison with literature data. Hence they state that the effect of shading adds up to desiccation. In my opinion, the additive effect is not clear. It looks like the difference between light and shaded plants (see RGR) decreases with increasing exposure and is probably not significant at 6-8hr.
In line 267 it is said that “shading further reduced the negative effect of emergence”. Is this what authors wanted to say? If this is the case, low light counteracts the negative effects of desiccation. This is what should theoretically happen, but it contrasts with the main conclusions driven in the paper.
According to the fact the low light is not a clear stressing factor for the upper limit of plants and is not fully appropriate to be tested together with desiccation, in the conclusive paragraphs most of the speculation is about the effect of low light in determining the depth distribution of the species. Although this is surely interesting, it is not really in line with the goal of the work.
Line 247 and 266: conforming our hypothesis…
Line 268: the additive effect is not clear. It looks like the difference between light and shaded plants decreases with increasing exposure and is probably not significant at 6-8hr.

---

## Round 0.2 · Minor Revisions

ln 31: you removed PSU, as the reviewer requested, but now it is unclear what the numbers are reporting. you need to give some unit or description or label to interpret the meaning of "25, 30, and 35". you should also state the unit (ppt or g/kg, I guess) the first time you refer to salinity values in other sections of the manuscript (e.g., ln 144, ln 202, in the Figure 2 caption, and so on).

Figure 2 caption: "on" should read "one" in the last sentence. there also appears to be a period missing between the second to last and the last sentence.

---

## Round 0.3 · accepted · Accept

I am satisfied with your rebuttal. Congratulations!

#